# Autonomous Expenditure Multipliers and Gross Value Added

**Arkadiusz J. Derkacz** 

Institute of Economics, University of Social Sciences, 00-842 Warsaw, Poland; aderkacz@san.edu.pl

**Abstract:** The paper aims to answer two main questions. Is it possible to calculate and analyze fiscal, investment and export multipliers in the short term? The classic approach is mainly based on the input–output balances, which are most often published every 5 years. Is it possible to determine the impact of autonomous expenditure on the growth rate of gross value added? Research and analysis are based primarily on the principle of aggregate demand and the main assumptions of the economic Keynesian model. In the paper, I present theoretical considerations to answer research questions. I have verified the proposed method for calculating the multipliers of autonomous expenditure and the relationship between autonomous expenditure and gross value added in empirical studies. To this end, I have chosen the three economies of the Weimar Group countries. It has emerged that the proposed method allows us to examine the growth rate of value added relative to *GDP* in the short term, while using the fiscal, investment and export multipliers mechanism.

**Keywords:** principle of efficient demand; fiscal multiplier; investment multiplier; export multiplier; GVA; development economy

**JEL Classification:** E0; E12; E20; E63

## 1. Introduction

Analysis of economic literature makes it possible to conclude that the issue of autonomous expenditure multipliers is one of the most important in modern economic development. Autonomous expenditure appears in contemporary literature in various contexts. Allain describes the Kalecki-Harrodian model which shows that the usual short-run properties are only transient, since the long-run growth rate converges towards that of autonomous expenditures. However, the impact on the level of variables (output, capital stock, labor, etc.) is permanent (Allain 2015). You can also meet the term "the economics of the super-multiplier", which Palley refers to in the labor market (Palley 2019). The chapter by Sawyer, in which he discusses the idea of the multiplier by Kalecki, is also interesting. It is interpreted here as a general idea according to which investments generate changes in output and income (Sawyer 2008). Studies on the value of multipliers of autonomous expenditure, e.g., in the context of economic crises, also often appear in the literature (Naimzada and Pecora 2017; Pusch 2012). Various methods are also used to estimate autonomous expenditure multipliers, such as the panel instrumental-variables approach (Girardi and Pariboni 2015). In this article, however, I adopt the concept of multipliers according to the theory of aggregate demand. They reveal how *GDP* responds to fiscal, investment and export policies in capitalist economies. However, there are some doubts about the methodology for estimating multipliers. They concern the method of calculating import-intensity coefficients. Moreover, modern economics, above all development economics and social economy, strongly emphasizes the predominance of the gross value-added ratio over gross domestic product. It is the gross value added (*GVA*) that determines the value of all the socio-economic benefits that are generated in the process of producing goods and services in the economy.

In this context, I have attempted to answer two key questions. The first concerns how the import capacity of autonomous expenditure is calculated. Is it possible to estimate import-intensive ratios—thus autonomous expenditure multipliers—in the short term? In various publications describing the multipliers of autonomous expenditure, information appears about a certain methodological difficulty. It concerns the calculation of the import intensity of domestic accumulation (Łaski et al. 2010b). Very often, these publications use the input–output balance (Rueda-Cantuche et al. 2017; Statistics Poland 2019). It is published every five years. This is why the idea of a different approach to the calculation of import intensity indicators appeared. The second question concerns gross value added and reads as follows: Is it possible to estimate changes in gross value added as a result of changes in autonomous expenditure? This problem is justified by the approach to calculating the *GDP* value. According to the theory of total demand, *GDP* is most often analyzed in terms of its distribution or income. In his research, however, I emphasizes *GDP* in terms of production. This is why an attempt has been made here to transform the classical formula of *GDP* dynamics, which is used in the theory of aggregate demand.

The question raised in this way is determined by two main objectives. First, I tried to redefine the method for calculating the import-absorbency ratios of autonomous expenditure. I used the breakdown of imports here according to Broad Economic Categories (BEC). As a result, I have divided total imports into consumer, capital and supply goods. In addition, I have proposed the introduction of appropriate factors that determine the share of government expenditure and exports in gross value added. This allowed the imported goods to be divided between the different elements of final production. In addition, this procedure gave the opportunity to calculate the multipliers of autonomous expenditure in the short term. The second objective of the work is to determine the dynamics of the development of *GVA* in relation to gross domestic product. In this respect, I have attempted to verify the theoretical mechanism of autonomous expenditure multipliers. I was mainly basing this on aggregate demand. In adopting the most important theoretical assumptions, I have attempted to transform a key formula that classically determines the level of *GDP* response to changes in autonomous expenditure. This has allowed the dynamics of changes in gross value added to be defined using fiscal, investment and export multiplier mechanisms.

I verified the proposed theoretical solutions in empirical studies. To this end, I have focused on the economies of the Weimar Group countries. They are France, Germany and Poland. The choice is dictated by the fact that the economies of these three countries are quite different. The aim of this choice of economies is to try to verify the method of calculating the multipliers of autonomous expenditure in different economic circumstances. I used Eurostat statistics for calculations and analyses. I accepted the period 2010–2019. All data used is quarterly data. This was to see if it would be possible to analyze the multipliers of autonomous expenditure in the short term. After presenting the most important theoretical assumptions, I present a proposal to decompose total imports in the economy and to recast the appropriate formulas according to the principle of aggregate demand. I verified the results in empirical studies.

Finally, I presented the most important conclusions that I drew from my analyses and studies. These are mainly the two main conclusions. The use of the proposed method for calculating fiscal, investment and export multipliers allows for a short-term analysis of the impact of autonomous expenditure on the development of the economy. Thus, it is possible to reduce the limitation of research and analysis, which is caused by the frequency of publication of the input–output balances. The second conclusion concerns how the mechanism for multipliers of autonomous expenditure is interpreted. Fiscal, investment and export multipliers have shown that in a real way, they determine the response of *GVA* growth to changes in autonomous expenditure. It can therefore be said that the proposed method of calculating the multipliers of autonomous expenditure can be valuable for further research and analysis and for economic policy practice.

## 2. Method and Economic Background

The analyses and studies presented in the paper are based on the theory of effective demand. I completely reject the assumptions of the mainstream economy, which speaks of the automatic pursuit of a free market economy to a state of equilibrium in full use of production capacity (Akerlof 2007; Dequech 2007). This approach coincides with the theory of effective demand, which is the main context of economic research for the author (Łaski 2019, pp. XXIX–XXXVI). The most important element of the theory of effective demand is the assumption that the economy is experiencing an inseparable state of instability (Akerlof and Yellen 1987). It is determined by investment decisions. The issue of underutilization of production resources in capitalist economies is also important. Changes in gross domestic product remain determined by the volume of autonomous expenditure. These include expenditure on private investment, public expenditure and exports. Private investment is defined here, in accordance with the System of National Accounts, as gross accumulation (World Bank 2009, pp. 198–207). The ratio of *GDP* change to changes in individual autonomous expenditure is defined as the multipliers of those expenditures. So, we have a multiplier of public expenditure, an investment multiplier and an export multiplier. I use the same method to estimate the value of multipliers, which is used in similar publications on the study of autonomous expenditure measures (Łaski et al. 2010c; Palley 2009). Thus, autonomous expenditure is a factor in the combined demand in the capitalist economy. The increase in autonomous expenditure increases the value of gross domestic product. On the other hand, there are leaks in aggregate demand (Łaski et al. 2012). Here, we are dealing with taxes, imports and private savings. This means that the growth of these volumes in the economy is slowing its growth.

In the classical approach, fiscal, investment and export multipliers show the extent to which *GDP* reacts to the change in individual autonomous expenditures. *GDP* change is not always equal to a change in the value of expenditure supplying total global demand. This is due to leaks of aggregate demand, which are disclosed in the values of multipliers. The examination of changes in the value of the fiscal, investment and export multipliers is therefore important for economic policy forecasting. The stable level of these multipliers in the long term stabilizes the economic policy environment. Uncertainty over the volatility of the multipliers of autonomous expenditure can be a source of economic risk (Andrade et al. 2018). The uncertainty of the value of the fiscal, investment and export multiplier is relevant for the corporate sector (Ko et al. 2018) and for the public sector (Fischer 2011; Schultz 2002). In this paper, I will try to verify the multiplier calculation process in such a way as to show the response of gross value added to changes in autonomous expenditure.

In my analyses, in addition to the principle of aggregate demand, I also adopted the basic assumptions of the Keynesian model. I therefore make the following assumptions. The economy is under-exploiting manufacturing factors. Foreign trade is balanced without many restrictions. There are slight changes in the distribution of national income between wages and gross profit margin. In the capitalist economy, an accommodative money supply policy is implemented. On this basis, gross domestic product is the sum of private consumption, *CP*, private investment, *IP*, public expenditure, *G*, and exports, *X*, minus the value of imports, *M* (Łaski 2019, pp. 1–5). This can be represented in the equation:

$$Y = CP + IP + G + X - M \tag{1}$$

I also accept that the sum of net taxes, *TN*, is the difference between total tax revenues and burdens on the corporate and household sector and all public sector transfers. On this basis, I determine the value of the disposable income of the private sector $YD = Y - TN$. At the same time, I define the *TN* value as public sector disposable income. We can save this as $Y = YD + TN$ and $YD = Y - TN$. At this point, I introduce the concept of the gross savings value of the private sector, *SP*. This is the difference between private sector disposable income and private consumption. We can save this as $CP = Y - TN - SP$ and $Y = CP + TN + SP$.

I also introduce further considerations as to the concept of the average private savings rate, $sp = \frac{SP}{Y}$, and the average net tax rate, $tn = \frac{TN}{Y}$ (Łaski 2019, pp. 36, 59). After converting the last 3 formulas, I get an equation that determines the level of private consumption, $CP = (1 - tn - sp)Y$. The value of $cp = (1 - tn - sp)$ indicates the propensity for private consumption in the economy. On this basis, we get a formula:

$$CP = cpY \tag{2}$$

### 2.1. Import Decomposition by BEC

In the classic approach of the analysis of the multipliers of autonomous expenditure, the issue of decomposition of imports arises at this point. It is based primarily on domestic absorption and final production. The input–output balances are also used to estimate the value of multipliers (De March 2008; Rueda-Cantuche et al. 2017; Statistics Poland 2019). It should also be mentioned that various publications describing the multipliers of autonomous expenditure show some methodological difficulty. It concerns the calculation of the national accumulation import absorbance index (Łaski et al. 2010b, p. 17). In this paper, however, I propose a different approach to decomposition of imports. To do this, I used Broad Economic Categories (BEC) (United Nations 2002). On this basis, I have divided imports of goods into imports of intermediate goods, $M_{DZ}$, capital goods, $M_{DI}$, and consumption goods, $M_{DK}$. Total imports are as follows:

$$M = M_{DZ} + M_{DI} + M_{DK} \tag{3}$$

Such a divided import can be attributed to the main expenditure supplying the aggregate demand, *IP*, *CP*, *G*, *X* (see Figure 1). The entire import of consumption goods is absorbed by private consumption, *CP*. Imports of investment goods are also fully absorbed, but through gross *IP* accumulation. Gross accumulation, in addition to capital goods, also absorbs some of the imported intermediate goods, $M_{DZ}$. Imported intermediates are absorbed in the economy by gross capital formation, public expenditure (including public investment) and exports. In order to break down the imports of intermediate goods, I introduce certain coefficients.

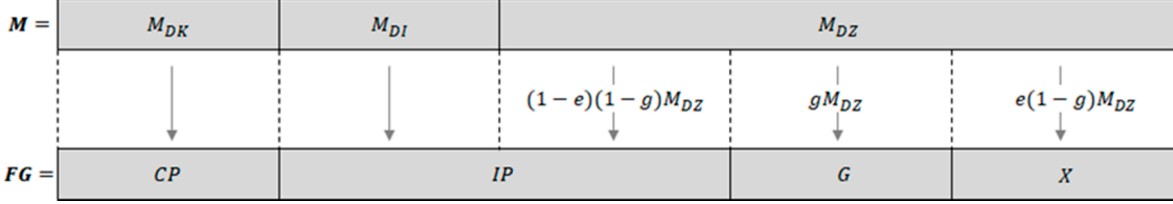

**Figure 1.** Decomposition of import and final production, own work.

I define public expenditure, *G*, as the value of consumption in the public sector. Some of them are used to obtain imported intermediate goods. Therefore, I introduce here the factor $g$ ($0 < g < 1$). It means the share of public expenditure in gross value added in a given period. In this way, the state activity disclosed in public expenditure absorbs the import of intermediate goods with a value of $gM_{DZ}$. So, there remains the second part of the import of intermediate goods worth $(1 - g)M_{DZ}$, which is absorbed by the enterprise sector. I define this part of the import of semi-finished products as $pM_{DZ}$, where $p = 1 - g$. In this way, I split the import of intermediate goods between the private and public sectors.

It is also necessary to divide the import of intermediate goods absorbed by enterprises ($pM_{DZ}$) into domestic accumulation and export. For this, I introduce the coefficient $e$ ($0 < e < 1$). It means the share of exports in *GVA* in a given period. Using this coefficient, I calculate the part of the import of intermediate goods that is used to produce goods intended for export. This value can be specified as $epM_{DZ}$. The remaining part of imported intermediate goods is absorbed by private enterprises for

domestic absorption. I present this value as $(1-e)pM_{DZ}$. I also introduce the coefficient $a = 1-e$. It determines the share of the part of intermediate goods imports absorbed for the purposes of gross capital formation ($apM_{DZ}$).

On this basis, the total import decomposition is:

$$M = M_{DK} + (M_{DI} + apM_{DZ}) + gM_{DZ} + epM_X \tag{4}$$

Next, I define the individual levels of import intensity. The import intensity of the national accumulation is determined by the part of the import $M_A = M_{DK} + (M_{DI} + apM_{DZ}) + gM_{DZ}$. Import intensity of the final production is additionally dependent on the value of $epM_X$. On this basis, I determine the import intensity coefficients for individual autonomous expenditure:

$$m_{CP} = \frac{M_{DK}}{CP}; \; m_G = \frac{gM_{DZ}}{G}; \; m_{IP} = \frac{M_{DI} + apM_{DZ}}{IP}; \; m_X = \frac{epM_{DZ}}{X} \tag{5}$$

This way of interpreting total imports is marked by the yoke of approximate estimates. However, I believe that such a solution can be valuable for a short-term analysis of import intensity and fiscal, investment and export multipliers. The very fact that I gave up the input–output balances in favor of decomposition of imports according to BEC makes it possible to study the variability of autonomous expenditure multipliers in the short term. Input–output balances are published every 5 years. In contrast, data on international trade are quarterly.

## 2.2. Autonomous Expenditure Multipliers

Based on the above considerations, I will try to derive the formulas for the fiscal, investment and export multipliers. For this purpose, I transform Equation (1). Using Equations (2) and (5), we have:

$$Y = cpY + IP + G + X - m_{CP}cpY - m_{IP}IP - m_G G - m_X X$$
$$Y = cpY(1 - m_{CP}) + IP(1 - m_{IP}) + G(1 - m_G) + X(1 - m_X)$$

At this point, I am assuming the value of *GDP* by production approach. This means that:

$$Y = GVA + TN \tag{6}$$

where *GVA* is gross value added. I continue the above transformations by introducing the Equation (6):

$$GVA + TN = cp(GVA + TN)(1 - m_{CP}) + IP(1 - m_{IP}) + G(1 - m_G) + X(1 - m_X)$$
$$(GVA + TN) - cp(GVA + TN)(1 - m_{CP}) = IP(1 - m_{IP}) + G(1 - m_G) + X(1 - m_X)$$
$$(GVA + TN)(1 - cp(1 - m_{CP})) = IP(1 - m_{IP}) + G(1 - m_G) + X(1 - m_X)$$

Finally, I get the following equation:

$$GVA = \frac{IP(1 - m_{IP}) + G(1 - m_G) + X(1 - m_X)}{1 - cp(1 - m_{CP})} - TN \tag{7}$$

Based on this equation, I can distinguish the fiscal ($k_1$), investment ($k_2$) and export ($k_3$) multipliers:

$$k_1 = \frac{(1 - m_G)}{1 - cp(1 - m_{CP})}; \; k_2 = \frac{(1 - m_{IP})}{1 - cp(1 - m_{CP})}; \; k_3 = \frac{(1 - m_X)}{1 - cp(1 - m_{CP})} \tag{8}$$

I write a simplified version of Equation (7) as:

$$GVA = k_1 G + k_2 IP + k_3 X - TN \tag{9}$$

It follows that the value *GVA* is completely dependent on the value of autonomous expenditure, on the coefficients, *k*, and the value of taxes, *TN*.

Traditionally, the fiscal, investment and export multipliers determine how much *GDP* changes as a result of changes in individual autonomous expenditure. In the proposed approach, I obtained a different relationship. Assuming a given tax value, we see how the level of *GVA* responds to the level of autonomous expenditure. Assuming that the multipliers *k* are given, I can analyze the impact of changes in autonomous expenditure on the change in gross value added (Łaski et al. 2010a, p. 810). For this purpose, Equation (7) takes the following form:

$$\Delta GVA = \frac{\Delta IP(1 - m_{IP}) + \Delta G(1 - m_G) + \Delta X(1 - m_X)}{1 - cp(1 - m_{CP})} - \Delta TN \tag{10}$$

To analyze the above relationship in terms of *GDP*, I divide both sides of the equation by *Y*:

$$\frac{\Delta GVA}{Y} = \frac{1}{Y}\frac{\Delta IP(1 - m_{IP}) + \Delta G(1 - m_G) + \Delta X(1 - m_X)}{1 - cp(1 - m_{CP})} - \frac{\Delta TN}{Y}$$

I denote $r_{GVA} = \frac{\Delta GVA}{Y}$. This value means the ratio of changes in gross value added to *GDP*. Thus, the value $r_{GVA}$ is defined as the growth dynamics of *GVA* in relation to gross domestic product. I write the final form of the main equation as:

$$r_{GVA} = k_1\frac{\Delta G}{Y} + k_2\frac{\Delta IP}{Y} + k_3\frac{\Delta X}{Y} - \Delta tn \tag{11}$$

It follows that the dynamics of gross value added growth in the capitalist open economy depends on:

- Changes in the share of individual autonomous expenditure in *GDP*,
- Changes in the average tax rate and
- The values of the fiscal, investment and export multipliers.

I interpret this equation as follows. The growth rate of *GVA* depends on two phenomena. The first is the so-called injection of autonomous expenditure. The increase in these expenditures accelerates the growth of gross value added in the economy. However, it is adjusted by the value of the multipliers and the change in the average taxation. This is where a second phenomenon emerges, which I refer to as leakage. The classic approach talks about leaks in aggregate demand. It is determined by the value of the fiscal, investment and export multipliers. They are mainly determined by the $1 - cp(1 - m_{CP})$ value. This means that the change in gross value added reacts to the change in autonomous expenditure depending on the level of propensity to private consumption and the level of import intensity of private consumption. The import intensity of autonomous expenditure is also important. These values can be found in the nominator of the individual multipliers. In Equation (11), we see one more leak that slows down the dynamics of gross value added growth. It is the change in average taxation ($\Delta tn$). I call it a domestic leak. This means that the increase in taxation in the economy has a negative impact on the growth dynamics of gross value added.

In the above considerations, I have presented a proposal for the decomposition of imports according to the main economic categories of BEC. I also tried to describe the theoretical mechanisms of the functioning of the fiscal, investment and export multipliers. Appropriate transformations of the formulas used in the traditional approach allowed to determine the impact of changes in autonomous expenditure on the dynamics of the growth of gross value added. In the next part of the paper, I will attempt to estimate the multipliers of autonomous expenditure and to analyze the dynamics of gross value added growth.

## 3. Results of Calculating Autonomous Expenditure Multipliers

In this section, I present the calculations to verify the theoretical issues discussed above. I chose the Weimar Triangle countries for statistical analysis. They are France, Germany and Poland. If the proposed method of testing autonomous expenditure multipliers is verified positively, detailed studies of other countries will be possible. All calculations and analyses are based on Eurostat statistics. For the analysis, I used the period of the last nine years (2010–2019). Thanks to the BEC division of imports, I was able to use quarterly data. Thus, all analyses should be treated as short-term analyses.

### 3.1. Import-Intensity Coefficients

First, I calculated the share of individual groups of imported goods in total imports. For this purpose, I used the international trade data of the EU Member States according to BEC. In Table 1, I present the share of imports of capital goods in total imports. Table 2 contains data on the share of imports of consumption goods in total imports. However, the share of imports of intermediate goods in total imports is presented in Table 3.

**Table 1.** Share of imports of capital goods in total imports (in %), own work, source: Eurostat.

| | 2010 Q1 | 2010 Q2 | 2010 Q3 | 2010 Q4 | 2011 Q1 | 2011 Q2 | 2011 Q3 | 2011 Q4 | 2012 Q1 | 2012 Q2 | 2012 Q3 | 2012 Q4 | 2013 Q1 | 2013 Q2 | 2013 Q3 | 2013 Q4 | 2014 Q1 | 2014 Q2 | 2014 Q3 | 2014 Q4 |
|---|---|---|---|---|---|---|---|---|---|---|---|---|---|---|---|---|---|---|---|---|
| FRA | 0.14 | 0.15 | 0.14 | 0.16 | 0.14 | 0.14 | 0.15 | 0.15 | 0.14 | 0.15 | 0.14 | 0.15 | 0.14 | 0.14 | 0.14 | 0.15 | 0.14 | 0.14 | 0.14 | 0.15 |
| GER | 0.15 | 0.16 | 0.15 | 0.17 | 0.14 | 0.15 | 0.14 | 0.16 | 0.14 | 0.14 | 0.15 | 0.15 | 0.14 | 0.15 | 0.14 | 0.16 | 0.15 | 0.15 | 0.14 | 0.16 |
| POL | 0.17 | 0.17 | 0.16 | 0.18 | 0.15 | 0.16 | 0.17 | 0.17 | 0.15 | 0.16 | 0.17 | 0.17 | 0.15 | 0.18 | 0.16 | 0.18 | 0.15 | 0.17 | 0.16 | 0.18 |

| | 2015 Q1 | 2015 Q2 | 2015 Q3 | 2015 Q4 | 2016 Q1 | 2016 Q2 | 2016 Q3 | 2016 Q4 | 2017 Q1 | 2017 Q2 | 2017 Q3 | 2017 Q4 | 2018 Q1 | 2018 Q2 | 2018 Q3 | 2018 Q4 | 2019 Q1 | 2019 Q2 | 2019 Q3 | 2019 Q4 |
|---|---|---|---|---|---|---|---|---|---|---|---|---|---|---|---|---|---|---|---|---|
| FRA | 0.15 | 0.15 | 0.15 | 0.16 | 0.16 | 0.16 | 0.15 | 0.17 | 0.14 | 0.15 | 0.15 | 0.15 | 0.14 | 0.14 | 0.14 | 0.15 | 0.15 | 0.15 | 0.15 | 0.16 |
| GER | 0.16 | 0.15 | 0.15 | 0.17 | 0.16 | 0.16 | 0.16 | 0.17 | 0.16 | 0.15 | 0.15 | 0.16 | 0.15 | 0.15 | 0.15 | 0.17 | 0.16 | 0.15 | 0.15 | 0.17 |
| POL | 0.17 | 0.18 | 0.18 | 0.20 | 0.18 | 0.18 | 0.16 | 0.17 | 0.15 | 0.17 | 0.16 | 0.17 | 0.16 | 0.17 | 0.17 | 0.17 | 0.17 | 0.17 | 0.16 | 0.18 |

**Table 2.** Share of imports of consumption goods in total imports (in %), own work, source: Eurostat.

| | 2010 Q1 | 2010 Q2 | 2010 Q3 | 2010 Q4 | 2011 Q1 | 2011 Q2 | 2011 Q3 | 2011 Q4 | 2012 Q1 | 2012 Q2 | 2012 Q3 | 2012 Q4 | 2013 Q1 | 2013 Q2 | 2013 Q3 | 2013 Q4 | 2014 Q1 | 2014 Q2 | 2014 Q3 | 2014 Q4 |
|---|---|---|---|---|---|---|---|---|---|---|---|---|---|---|---|---|---|---|---|---|
| FRA | 0.31 | 0.29 | 0.30 | 0.30 | 0.28 | 0.28 | 0.29 | 0.29 | 0.27 | 0.27 | 0.28 | 0.29 | 0.28 | 0.28 | 0.29 | 0.29 | 0.29 | 0.29 | 0.31 | 0.31 |
| GER | 0.25 | 0.23 | 0.24 | 0.23 | 0.23 | 0.22 | 0.23 | 0.22 | 0.23 | 0.22 | 0.24 | 0.23 | 0.23 | 0.22 | 0.24 | 0.24 | 0.24 | 0.24 | 0.25 | 0.25 |
| POL | 0.23 | 0.21 | 0.22 | 0.22 | 0.22 | 0.19 | 0.20 | 0.21 | 0.21 | 0.20 | 0.20 | 0.21 | 0.22 | 0.21 | 0.21 | 0.22 | 0.23 | 0.21 | 0.22 | 0.23 |

| | 2015 Q1 | 2015 Q2 | 2015 Q3 | 2015 Q4 | 2016 Q1 | 2016 Q2 | 2016 Q3 | 2016 Q4 | 2017 Q1 | 2017 Q2 | 2017 Q3 | 2017 Q4 | 2018 Q1 | 2018 Q2 | 2018 Q3 | 2018 Q4 | 2019 Q1 | 2019 Q2 | 2019 Q3 | 2019 Q4 |
|---|---|---|---|---|---|---|---|---|---|---|---|---|---|---|---|---|---|---|---|---|
| FRA | 0.31 | 0.31 | 0.33 | 0.32 | 0.33 | 0.33 | 0.34 | 0.33 | 0.32 | 0.32 | 0.33 | 0.32 | 0.32 | 0.32 | 0.32 | 0.32 | 0.31 | 0.32 | 0.33 | 0.34 |
| GER | 0.26 | 0.25 | 0.27 | 0.27 | 0.27 | 0.27 | 0.28 | 0.28 | 0.26 | 0.27 | 0.27 | 0.27 | 0.26 | 0.26 | 0.26 | 0.27 | 0.26 | 0.27 | 0.28 | 0.29 |
| POL | 0.25 | 0.22 | 0.24 | 0.25 | 0.26 | 0.25 | 0.26 | 0.27 | 0.26 | 0.25 | 0.26 | 0.28 | 0.27 | 0.25 | 0.26 | 0.27 | 0.26 | 0.26 | 0.28 | 0.28 |

**Table 3.** Share of imports of intermediate goods in total imports (in %), own work, source: Eurostat.

| | 2010 Q1 | 2010 Q2 | 2010 Q3 | 2010 Q4 | 2011 Q1 | 2011 Q2 | 2011 Q3 | 2011 Q4 | 2012 Q1 | 2012 Q2 | 2012 Q3 | 2012 Q4 | 2013 Q1 | 2013 Q2 | 2013 Q3 | 2013 Q4 | 2014 Q1 | 2014 Q2 | 2014 Q3 | 2014 Q4 |
|---|---|---|---|---|---|---|---|---|---|---|---|---|---|---|---|---|---|---|---|---|
| FRA | 0.55 | 0.56 | 0.55 | 0.54 | 0.58 | 0.58 | 0.57 | 0.56 | 0.59 | 0.57 | 0.58 | 0.56 | 0.58 | 0.58 | 0.56 | 0.55 | 0.57 | 0.56 | 0.55 | 0.54 |
| GER | 0.60 | 0.62 | 0.61 | 0.60 | 0.63 | 0.63 | 0.64 | 0.62 | 0.63 | 0.63 | 0.62 | 0.62 | 0.62 | 0.63 | 0.62 | 0.60 | 0.62 | 0.61 | 0.61 | 0.59 |
| POL | 0.60 | 0.62 | 0.62 | 0.60 | 0.63 | 0.64 | 0.63 | 0.63 | 0.64 | 0.64 | 0.63 | 0.62 | 0.62 | 0.62 | 0.62 | 0.60 | 0.62 | 0.62 | 0.62 | 0.59 |

| | 2015 Q1 | 2015 Q2 | 2015 Q3 | 2015 Q4 | 2016 Q1 | 2016 Q2 | 2016 Q3 | 2016 Q4 | 2017 Q1 | 2017 Q2 | 2017 Q3 | 2017 Q4 | 2018 Q1 | 2018 Q2 | 2018 Q3 | 2018 Q4 | 2019 Q1 | 2019 Q2 | 2019 Q3 | 2019 Q4 |
|---|---|---|---|---|---|---|---|---|---|---|---|---|---|---|---|---|---|---|---|---|
| FRA | 0.54 | 0.54 | 0.52 | 0.51 | 0.51 | 0.51 | 0.51 | 0.51 | 0.54 | 0.53 | 0.52 | 0.52 | 0.54 | 0.54 | 0.54 | 0.53 | 0.54 | 0.53 | 0.52 | 0.51 |
| GER | 0.58 | 0.60 | 0.58 | 0.56 | 0.57 | 0.57 | 0.57 | 0.56 | 0.58 | 0.58 | 0.58 | 0.57 | 0.59 | 0.59 | 0.59 | 0.57 | 0.58 | 0.58 | 0.56 | 0.55 |
| POL | 0.58 | 0.59 | 0.58 | 0.55 | 0.56 | 0.57 | 0.58 | 0.56 | 0.58 | 0.58 | 0.58 | 0.56 | 0.57 | 0.58 | 0.58 | 0.56 | 0.57 | 0.57 | 0.56 | 0.54 |

Based on the BEC division of imports, I estimated the proposed coefficients $g$, $p$, $e$ and $a$. It should be remembered that $p = 1 - g$ and $a = 1 - e$. Figure 2 shows the share of government expenditure in gross value added for the analyzed countries. These are the values of the factor $g$ which is used to calculate import-intensity levels. It turns out that the value of this coefficient is the highest in France and the lowest in Poland. I also calculated the coefficient of variation understood as the ratio of the standard deviation to the mean value of the variable under study (Abdi 2010, pp. 169–71). It turns

out that the value of this coefficient is relatively small. In the analyzed period, the *g* factor changes (coefficient of variation as the ratio of standard deviation to the average value 2010–2019) in France by 1.07%, in Germany by 1.77% and in Poland by 2.43%. On this basis, it can be said that the factor *g* is quite stable over the analyzed period. The next figure shows the calculated coefficient *e*. It denotes the share of total exports in gross value added (see Figure 3). The situation is slightly different here. The highest values of this coefficient occur in Poland, while the lowest in France. The factor *e* is less stable than the *g* factor. The share of exports in *GVA* changes by 7.39% in France, by 5.88% in Germany and by 12.95% in Poland. However, this does not change the fact that the coefficient *e* can be considered relatively stable in the short term.

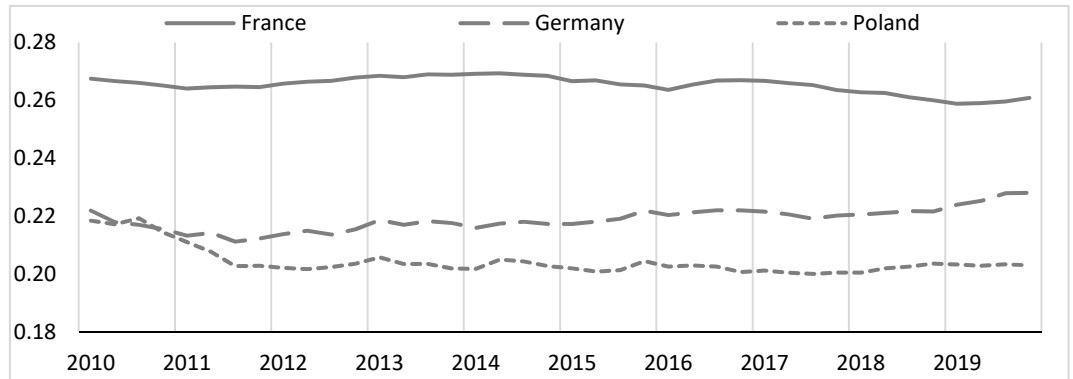

**Figure 2.** Government expenditure share in gross value added (factor g), own work, source: Eurostat.

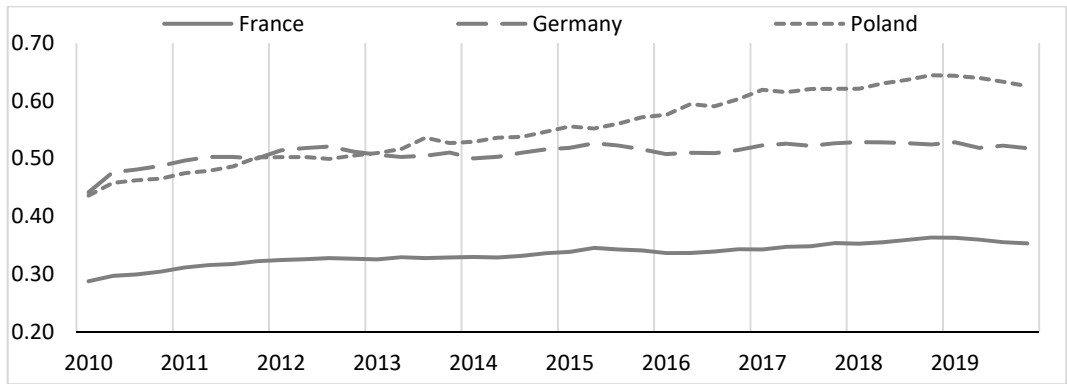

**Figure 3.** Export share in gross value added (factor e), own work, source: Eurostat.

Based on the data prepared in this way, I was able to calculate the values of the multipliers of autonomous expenditure according to Equation (5). Figure 4 presents the results of calculations of the private consumption import intensity. The value of this ratio varies to some extent in the analyzed period. The coefficient of variation for France is 6.28%, for Germany it is 9.03% and for Poland 19.67%. Additionally, an upward trend can be seen here, which means a short-term increase in the import intensity of private consumption. Assuming the base period of Q1 2010, the private consumption import intensity increased (in 4Q 2019) in France by 23.56%, in Germany by 32.52% and in Poland by 65.85%.

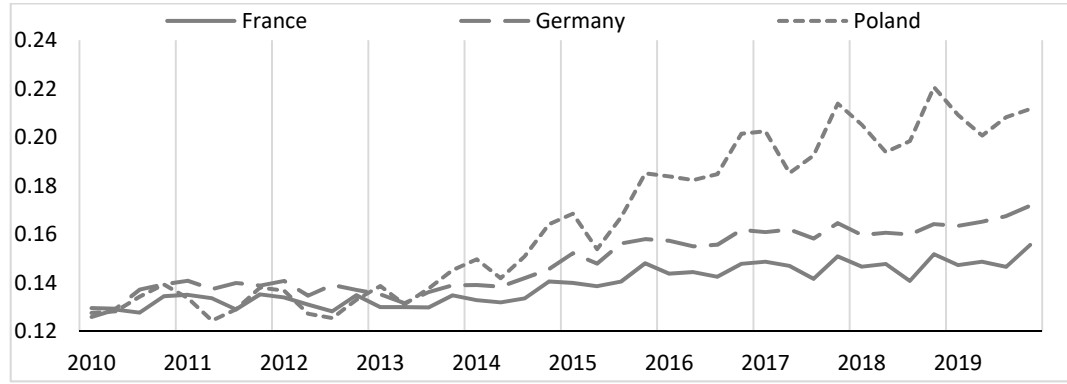

**Figure 4.** Import intensity of private consumption, own work, source: Eurostat.

Next, I calculated the import intensity of government expenditure. I present them in Figure 5. On this basis, it can be said that government expenditure on imported intermediate goods is the highest in Poland, and the lowest in France. The coefficient of variation for France is 8.38%, for Germany 7.93% and for Poland 6.95%. In the 1Q 2010 base period, the import intensity of government expenditure decreased by −1.09% in Germany. This ratio increased by 0.56% in France and by 12.68% in Poland.

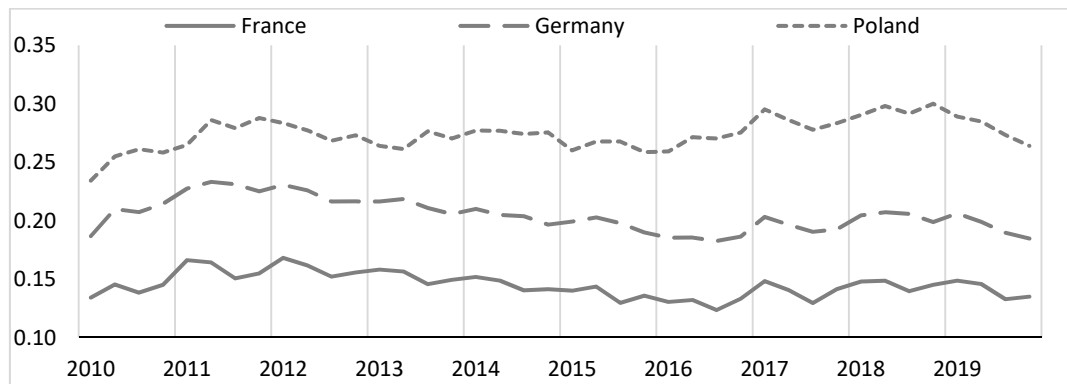

**Figure 5.** Import intensity of government expenditure, own work, source: Eurostat.

I also calculated the import intensity of gross capital formation. The results are presented in Figure 6. Also, in this situation, the highest level of import intensity is in Poland and the lowest in France. Additionally, the variability of this coefficient is relatively small. The coefficient of variation in the import intensity of gross accumulation is 7.22% in France, 5.58% in Germany and 5.06% in Poland. Taking the base period for Q1 2010 again, this import-intensity ratio declined in France (−12.87%) and Germany (−11.64%) in Q4 2019. In Poland, the change during this period was positive (8.07%).

I was the last to calculate the import intensity of export ratio. The results are presented in Figure 7. In this case, a fairly stable situation in the value of import intensity of export can also be noted. The coefficient of variation in France is 8.39%, in Germany 8.33% and in Poland 7.29%. In the analyzed period, the value of this ratio declines again to the base period (Q1 2010) in Germany (−1.87%). In France, in Q4 2019, the import intensity of exports increased slightly by 1.47% and in Poland by 14.91%.

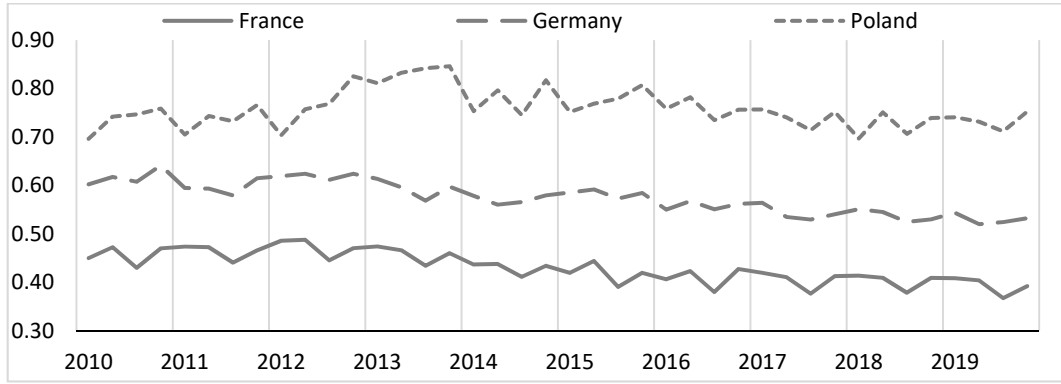

**Figure 6.** Import intensity of private investment, own work, source: Eurostat.

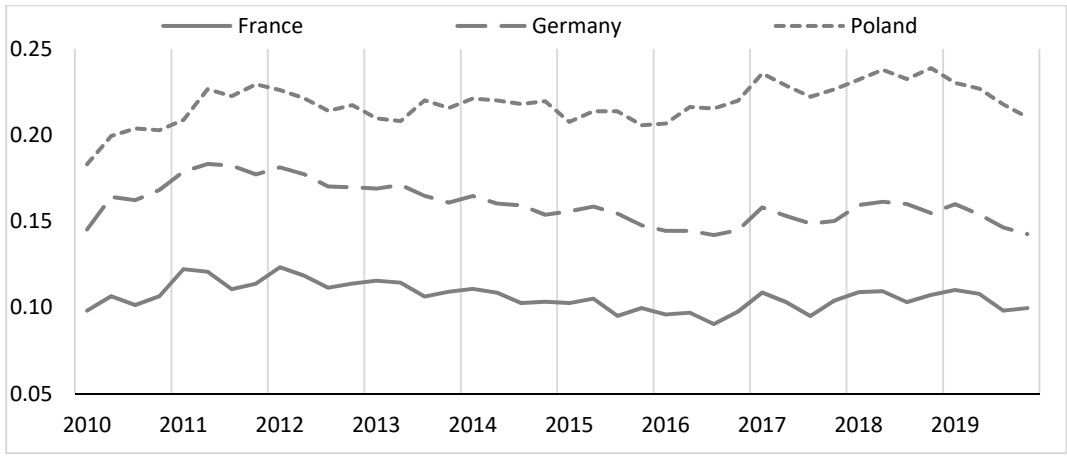

**Figure 7.** Import intensity of export.

### 3.2. Autonomous Expenditure Multipliers

The calculation of the fiscal, investment and export multipliers also requires the calculation of relevant indicators (see Equations (2) and (8)). I first calculated the private consumption propensity index. I used data on household final consumption expenditure. The results of the calculations are presented in Table 4. Based on these calculations, the values of the *CP* index are very stable in the analyzed period. The coefficient of variation is 1.18% for France, 2.37% for Germany and 2.71% for Poland.

**Table 4.** Propensity to private consumption—indicator *CP*, own work, source: Eurostat.

| | 2010 Q1 | 2010 Q2 | 2010 Q3 | 2010 Q4 | 2011 Q1 | 2011 Q2 | 2011 Q3 | 2011 Q4 | 2012 Q1 | 2012 Q2 | 2012 Q3 | 2012 Q4 | 2013 Q1 | 2013 Q2 | 2013 Q3 | 2013 Q4 | 2014 Q1 | 2014 Q2 | 2014 Q3 | 2014 Q4 |
|---|---|---|---|---|---|---|---|---|---|---|---|---|---|---|---|---|---|---|---|---|
| FRA | 0.54 | 0.53 | 0.53 | 0.53 | 0.54 | 0.53 | 0.53 | 0.53 | 0.53 | 0.53 | 0.52 | 0.53 | 0.53 | 0.52 | 0.53 | 0.53 | 0.52 | 0.52 | 0.52 | 0.52 |
| GER | 0.54 | 0.54 | 0.53 | 0.53 | 0.53 | 0.53 | 0.53 | 0.53 | 0.53 | 0.53 | 0.53 | 0.53 | 0.54 | 0.53 | 0.53 | 0.53 | 0.52 | 0.52 | 0.52 | 0.52 |
| POL | 0.61 | 0.60 | 0.61 | 0.61 | 0.61 | 0.61 | 0.61 | 0.60 | 0.61 | 0.61 | 0.60 | 0.61 | 0.60 | 0.60 | 0.61 | 0.60 | 0.60 | 0.60 | 0.59 | 0.59 |
| | **2015 Q1** | **2015 Q2** | **2015 Q3** | **2015 Q4** | **2016 Q1** | **2016 Q2** | **2016 Q3** | **2016 Q4** | **2017 Q1** | **2017 Q2** | **2017 Q3** | **2017 Q4** | **2018 Q1** | **2018 Q2** | **2018 Q3** | **2018 Q4** | **2019 Q1** | **2019 Q2** | **2019 Q3** | **2019 Q4** |
| FRA | 0.52 | 0.52 | 0.52 | 0.52 | 0.52 | 0.52 | 0.52 | 0.52 | 0.52 | 0.52 | 0.52 | 0.52 | 0.52 | 0.52 | 0.52 | 0.52 | 0.52 | 0.52 | 0.52 | 0.52 |
| GER | 0.52 | 0.52 | 0.51 | 0.51 | 0.51 | 0.51 | 0.51 | 0.51 | 0.51 | 0.51 | 0.51 | 0.51 | 0.51 | 0.51 | 0.51 | 0.51 | 0.51 | 0.51 | 0.51 | 0.51 |
| POL | 0.58 | 0.58 | 0.58 | 0.57 | 0.58 | 0.58 | 0.58 | 0.58 | 0.58 | 0.58 | 0.57 | 0.58 | 0.57 | 0.57 | 0.57 | 0.57 | 0.57 | 0.57 | 0.57 | 0.55 |

Secondly, I calculated the private savings propensity index. Here, I used the data of private savings in households. The results are presented in Table 5. In this case, the *SP* index is very stable in France (2.58%) and Germany (3.75%). In Poland, the coefficient of variation of the *SP* index is 10.14%.

**Table 5.** Propensity to private savings—indicator *SP*, own work, source: Eurostat.

|  | 2010 Q1 | 2010 Q2 | 2010 Q3 | 2010 Q4 | 2011 Q1 | 2011 Q2 | 2011 Q3 | 2011 Q4 | 2012 Q1 | 2012 Q2 | 2012 Q3 | 2012 Q4 | 2013 Q1 | 2013 Q2 | 2013 Q3 | 2013 Q4 | 2014 Q1 | 2014 Q2 | 2014 Q3 | 2014 Q4 |
|---|---|---|---|---|---|---|---|---|---|---|---|---|---|---|---|---|---|---|---|---|
| FRA | 0.31 | 0.31 | 0.32 | 0.31 | 0.31 | 0.32 | 0.32 | 0.32 | 0.32 | 0.32 | 0.32 | 0.30 | 0.30 | 0.31 | 0.30 | 0.29 | 0.31 | 0.30 | 0.31 | 0.31 |
| GER | 0.37 | 0.38 | 0.37 | 0.38 | 0.39 | 0.39 | 0.40 | 0.39 | 0.39 | 0.38 | 0.38 | 0.37 | 0.36 | 0.38 | 0.38 | 0.38 | 0.38 | 0.39 | 0.39 | 0.39 |
| POL | 0.20 | 0.21 | 0.18 | 0.18 | 0.19 | 0.17 | 0.18 | 0.19 | 0.17 | 0.20 | 0.19 | 0.18 | 0.19 | 0.20 | 0.20 | 0.20 | 0.20 | 0.19 | 0.21 | 0.21 |

|  | 2015 Q1 | 2015 Q2 | 2015 Q3 | 2015 Q4 | 2016 Q1 | 2016 Q2 | 2016 Q3 | 2016 Q4 | 2017 Q1 | 2017 Q2 | 2017 Q3 | 2017 Q4 | 2018 Q1 | 2018 Q2 | 2018 Q3 | 2018 Q4 | 2019 Q1 | 2019 Q2 | 2019 Q3 | 2019 Q4 |
|---|---|---|---|---|---|---|---|---|---|---|---|---|---|---|---|---|---|---|---|---|
| FRA | 0.31 | 0.30 | 0.31 | 0.32 | 0.31 | 0.30 | 0.31 | 0.30 | 0.31 | 0.31 | 0.31 | 0.32 | 0.31 | 0.32 | 0.32 | 0.32 | 0.33 | 0.32 | 0.33 | 0.33 |
| GER | 0.40 | 0.40 | 0.41 | 0.40 | 0.40 | 0.41 | 0.40 | 0.40 | 0.41 | 0.40 | 0.41 | 0.41 | 0.41 | 0.41 | 0.41 | 0.41 | 0.41 | 0.41 | 0.41 | 0.41 |
| POL | 0.28 | 0.20 | 0.21 | 0.22 | 0.22 | 0.24 | 0.20 | 0.23 | 0.22 | 0.21 | 0.23 | 0.22 | 0.21 | 0.22 | 0.21 | 0.21 | 0.23 | 0.22 | 0.26 | 0.25 |

I was the last to calculate the average tax rate, *TN*, in the economy. I made my calculations based on the values of total taxes on products' less subsidies. The results are presented in Table 6. The results of the calculations show a relatively high stability of this indicator in the analyzed period. The coefficient of variation for France is 4.21%, for Germany 1.86% and for Poland 3.83%.

**Table 6.** Average tax rate—indicator *TN*, own work, source: Eurostat

|  | 2010 Q1 | 2010 Q2 | 2010 Q3 | 2010 Q4 | 2011 Q1 | 2011 Q2 | 2011 Q3 | 2011 Q4 | 2012 Q1 | 2012 Q2 | 2012 Q3 | 2012 Q4 | 2013 Q1 | 2013 Q2 | 2013 Q3 | 2013 Q4 | 2014 Q1 | 2014 Q2 | 2014 Q3 | 2014 Q4 |
|---|---|---|---|---|---|---|---|---|---|---|---|---|---|---|---|---|---|---|---|---|
| FRA | 0.10 | 0.10 | 0.10 | 0.10 | 0.10 | 0.10 | 0.10 | 0.10 | 0.10 | 0.10 | 0.10 | 0.10 | 0.10 | 0.10 | 0.10 | 0.10 | 0.10 | 0.10 | 0.10 | 0.10 |
| GER | 0.10 | 0.10 | 0.10 | 0.10 | 0.10 | 0.10 | 0.10 | 0.10 | 0.10 | 0.10 | 0.10 | 0.10 | 0.10 | 0.10 | 0.10 | 0.10 | 0.10 | 0.10 | 0.10 | 0.10 |
| POL | 0.12 | 0.12 | 0.12 | 0.12 | 0.12 | 0.12 | 0.12 | 0.12 | 0.12 | 0.11 | 0.11 | 0.11 | 0.11 | 0.11 | 0.11 | 0.11 | 0.12 | 0.11 | 0.11 | 0.11 |

|  | 2015 Q1 | 2015 Q2 | 2015 Q3 | 2015 Q4 | 2016 Q1 | 2016 Q2 | 2016 Q3 | 2016 Q4 | 2017 Q1 | 2017 Q2 | 2017 Q3 | 2017 Q4 | 2018 Q1 | 2018 Q2 | 2018 Q3 | 2018 Q4 | 2019 Q1 | 2019 Q2 | 2019 Q3 | 2019 Q4 |
|---|---|---|---|---|---|---|---|---|---|---|---|---|---|---|---|---|---|---|---|---|
| FRA | 0.10 | 0.11 | 0.11 | 0.11 | 0.11 | 0.11 | 0.11 | 0.11 | 0.11 | 0.11 | 0.11 | 0.11 | 0.11 | 0.11 | 0.11 | 0.11 | 0.11 | 0.11 | 0.11 | 0.11 |
| GER | 0.10 | 0.10 | 0.10 | 0.10 | 0.10 | 0.10 | 0.10 | 0.10 | 0.10 | 0.10 | 0.10 | 0.10 | 0.10 | 0.10 | 0.10 | 0.10 | 0.10 | 0.10 | 0.10 | 0.10 |
| POL | 0.11 | 0.11 | 0.11 | 0.11 | 0.11 | 0.12 | 0.12 | 0.12 | 0.12 | 0.12 | 0.12 | 0.12 | 0.12 | 0.12 | 0.12 | 0.13 | 0.12 | 0.12 | 0.12 | 0.12 |

Based on the previous calculations, I can proceed to estimating the value of the autonomous expenditure multipliers. I use for the calculations Equation (8) and the results of the *g*, *p*, *e* and *a* coefficients. Table 7 presents the estimated quarterly values of the fiscal multiplier. The value of the multiplier of government expenditure in the analyzed countries is very stable. The coefficient of variation in the analyzed period amounts to 2.18% for France, 3.55% for Germany and 7.84% for Poland. The value of this multiplier in Poland fell by −31.13% in Q4 2019 compared to the base period (Q1 2010).

**Table 7.** Fiscal multiplier, own work, source: Eurostat.

|  | 2010 Q1 | 2010 Q2 | 2010 Q3 | 2010 Q4 | 2011 Q1 | 2011 Q2 | 2011 Q3 | 2011 Q4 | 2012 Q1 | 2012 Q2 | 2012 Q3 | 2012 Q4 | 2013 Q1 | 2013 Q2 | 2013 Q3 | 2013 Q4 | 2014 Q1 | 2014 Q2 | 2014 Q3 | 2014 Q4 |
|---|---|---|---|---|---|---|---|---|---|---|---|---|---|---|---|---|---|---|---|---|
| FRA | 1.63 | 1.60 | 1.61 | 1.59 | 1.55 | 1.54 | 1.57 | 1.55 | 1.53 | 1.55 | 1.56 | 1.55 | 1.56 | 1.55 | 1.57 | 1.56 | 1.55 | 1.56 | 1.57 | 1.56 |
| GER | 1.53 | 1.49 | 1.47 | 1.45 | 1.42 | 1.41 | 1.41 | 1.43 | 1.42 | 1.44 | 1.44 | 1.45 | 1.46 | 1.45 | 1.45 | 1.45 | 1.43 | 1.44 | 1.44 | 1.44 |
| POL | 1.63 | 1.57 | 1.56 | 1.56 | 1.55 | 1.53 | 1.53 | 1.49 | 1.50 | 1.54 | 1.55 | 1.54 | 1.54 | 1.54 | 1.51 | 1.49 | 1.47 | 1.48 | 1.44 | 1.43 |

|  | 2015 Q1 | 2015 Q2 | 2015 Q3 | 2015 Q4 | 2016 Q1 | 2016 Q2 | 2016 Q3 | 2016 Q4 | 2017 Q1 | 2017 Q2 | 2017 Q3 | 2017 Q4 | 2018 Q1 | 2018 Q2 | 2018 Q3 | 2018 Q4 | 2019 Q1 | 2019 Q2 | 2019 Q3 | 2019 Q4 |
|---|---|---|---|---|---|---|---|---|---|---|---|---|---|---|---|---|---|---|---|---|
| FRA | 1.55 | 1.56 | 1.57 | 1.55 | 1.56 | 1.57 | 1.59 | 1.57 | 1.53 | 1.54 | 1.57 | 1.53 | 1.53 | 1.53 | 1.55 | 1.52 | 1.52 | 1.53 | 1.55 | 1.54 |
| GER | 1.42 | 1.42 | 1.42 | 1.43 | 1.43 | 1.43 | 1.44 | 1.43 | 1.40 | 1.40 | 1.41 | 1.39 | 1.38 | 1.37 | 1.38 | 1.38 | 1.37 | 1.39 | 1.40 | 1.40 |
| POL | 1.43 | 1.44 | 1.42 | 1.38 | 1.40 | 1.37 | 1.38 | 1.34 | 1.31 | 1.34 | 1.35 | 1.31 | 1.30 | 1.31 | 1.31 | 1.26 | 1.29 | 1.31 | 1.31 | 1.33 |

Then, I estimated the value of the investment multiplier (see Table 8). Its value for France and Germany is relatively stable. The coefficient of variation in the analyzed period amounts to 4.58% for France and 5.44% for Germany. In Poland, the investment multiplier is less stable in this period. The coefficient of variation is 16.28%.

**Table 8.** Investment multiplier, own work, source: Eurostat.

| | 2010 Q1 | 2010 Q2 | 2010 Q3 | 2010 Q4 | 2011 Q1 | 2011 Q2 | 2011 Q3 | 2011 Q4 | 2012 Q1 | 2012 Q2 | 2012 Q3 | 2012 Q4 | 2013 Q1 | 2013 Q2 | 2013 Q3 | 2013 Q4 | 2014 Q1 | 2014 Q2 | 2014 Q3 | 2014 Q4 |
|---|---|---|---|---|---|---|---|---|---|---|---|---|---|---|---|---|---|---|---|---|
| FRA | 1.03 | 0.99 | 1.07 | 0.99 | 0.98 | 0.98 | 1.04 | 0.98 | 0.95 | 0.95 | 1.02 | 0.97 | 0.97 | 0.98 | 1.04 | 0.99 | 1.03 | 1.03 | 1.08 | 1.03 |
| GER | 0.75 | 0.72 | 0.73 | 0.66 | 0.74 | 0.75 | 0.77 | 0.71 | 0.70 | 0.70 | 0.72 | 0.70 | 0.72 | 0.75 | 0.79 | 0.74 | 0.76 | 0.79 | 0.78 | 0.75 |
| POL | 0.65 | 0.55 | 0.53 | 0.51 | 0.62 | 0.55 | 0.57 | 0.49 | 0.62 | 0.52 | 0.49 | 0.37 | 0.39 | 0.35 | 0.33 | 0.31 | 0.50 | 0.42 | 0.51 | 0.36 |

| | 2015 Q1 | 2015 Q2 | 2015 Q3 | 2015 Q4 | 2016 Q1 | 2016 Q2 | 2016 Q3 | 2016 Q4 | 2017 Q1 | 2017 Q2 | 2017 Q3 | 2017 Q4 | 2018 Q1 | 2018 Q2 | 2018 Q3 | 2018 Q4 | 2019 Q1 | 2019 Q2 | 2019 Q3 | 2019 Q4 |
|---|---|---|---|---|---|---|---|---|---|---|---|---|---|---|---|---|---|---|---|---|
| FRA | 1.05 | 1.01 | 1.10 | 1.04 | 1.07 | 1.04 | 1.12 | 1.03 | 1.05 | 1.06 | 1.12 | 1.05 | 1.06 | 1.07 | 1.13 | 1.06 | 1.07 | 1.07 | 1.14 | 1.08 |
| GER | 0.75 | 0.74 | 0.77 | 0.74 | 0.79 | 0.77 | 0.80 | 0.78 | 0.75 | 0.81 | 0.82 | 0.78 | 0.79 | 0.80 | 0.83 | 0.81 | 0.80 | 0.84 | 0.82 | 0.81 |
| POL | 0.48 | 0.45 | 0.43 | 0.36 | 0.46 | 0.41 | 0.50 | 0.45 | 0.45 | 0.49 | 0.53 | 0.45 | 0.56 | 0.46 | 0.54 | 0.47 | 0.47 | 0.49 | 0.52 | 0.45 |

I was the last to calculate the export multiplier (see Table 9). The values of this multiplier have similar characteristics to the values of the fiscal multiplier. The coefficient of variation of the export multiplier in the analyzed period amounts to 1.97% for France, 3.32% for Germany and 7.46% for Poland. It should be noted that for each of the analyzed countries, the values of the export multiplier decreased in relation to the base period (Q1 2010). In France −5.50%, in Germany −8.23% and in Poland −18.31%.

**Table 9.** Export multiplier, own work, source: Eurostat.

| | 2010 Q1 | 2010 Q2 | 2010 Q3 | 2010 Q4 | 2011 Q1 | 2011 Q2 | 2011 Q3 | 2011 Q4 | 2012 Q1 | 2012 Q2 | 2012 Q3 | 2012 Q4 | 2013 Q1 | 2013 Q2 | 2013 Q3 | 2013 Q4 | 2014 Q1 | 2014 Q2 | 2014 Q3 | 2014 Q4 |
|---|---|---|---|---|---|---|---|---|---|---|---|---|---|---|---|---|---|---|---|---|
| FRA | 1.69 | 1.67 | 1.68 | 1.66 | 1.64 | 1.62 | 1.65 | 1.62 | 1.62 | 1.63 | 1.64 | 1.63 | 1.64 | 1.63 | 1.65 | 1.63 | 1.62 | 1.63 | 1.64 | 1.62 |
| GER | 1.61 | 1.57 | 1.55 | 1.54 | 1.51 | 1.50 | 1.50 | 1.51 | 1.51 | 1.53 | 1.53 | 1.54 | 1.55 | 1.54 | 1.54 | 1.54 | 1.51 | 1.52 | 1.52 | 1.52 |
| POL | 1.74 | 1.69 | 1.68 | 1.67 | 1.67 | 1.66 | 1.65 | 1.61 | 1.63 | 1.66 | 1.67 | 1.66 | 1.65 | 1.65 | 1.63 | 1.60 | 1.59 | 1.60 | 1.56 | 1.55 |

| | 2015 Q1 | 2015 Q2 | 2015 Q3 | 2015 Q4 | 2016 Q1 | 2016 Q2 | 2016 Q3 | 2016 Q4 | 2017 Q1 | 2017 Q2 | 2017 Q3 | 2017 Q4 | 2018 Q1 | 2018 Q2 | 2018 Q3 | 2018 Q4 | 2019 Q1 | 2019 Q2 | 2019 Q3 | 2019 Q4 |
|---|---|---|---|---|---|---|---|---|---|---|---|---|---|---|---|---|---|---|---|---|
| FRA | 1.62 | 1.63 | 1.64 | 1.61 | 1.62 | 1.63 | 1.64 | 1.63 | 1.61 | 1.61 | 1.63 | 1.60 | 1.60 | 1.60 | 1.62 | 1.59 | 1.59 | 1.59 | 1.61 | 1.60 |
| GER | 1.50 | 1.50 | 1.49 | 1.50 | 1.50 | 1.50 | 1.51 | 1.50 | 1.47 | 1.47 | 1.48 | 1.47 | 1.46 | 1.45 | 1.46 | 1.46 | 1.45 | 1.47 | 1.47 | 1.47 |
| POL | 1.53 | 1.55 | 1.52 | 1.48 | 1.50 | 1.48 | 1.49 | 1.45 | 1.42 | 1.45 | 1.45 | 1.42 | 1.41 | 1.42 | 1.42 | 1.37 | 1.39 | 1.41 | 1.41 | 1.42 |

*3.3. GVA Growth Dynamics Analysis*

The above calculations provide the basis for the analysis of the impact of autonomous expenditure on the dynamics of the growth of *GVA* in relation to *GDP*. For this purpose, I use Equation (11). I defined the dynamics of the growth of the gross added value $r_{GVA}$ as the ratio of the change in gross value added to the value of gross domestic product. To determine the change in the value of *GVA*, I assumed the absolute increases in the quarter compared to the corresponding quarter of the previous year $\Delta GVA = GVA(Q)_Y - GVA(Q)_{Y-1}$ (see Figure 8). These results show how much the *GVA* changed compared to the corresponding quarter of the previous year. On this basis, it is possible to analyze the absolute dynamics of gross value added in the short term. It is also possible to analyze the relative growth rate of *GVA* in relation to the *GDP* value. For this purpose, I use the relationship $r_{GVA} = \frac{\Delta GVA}{Y}$. The results for the analyzed countries are presented in Figure 9.

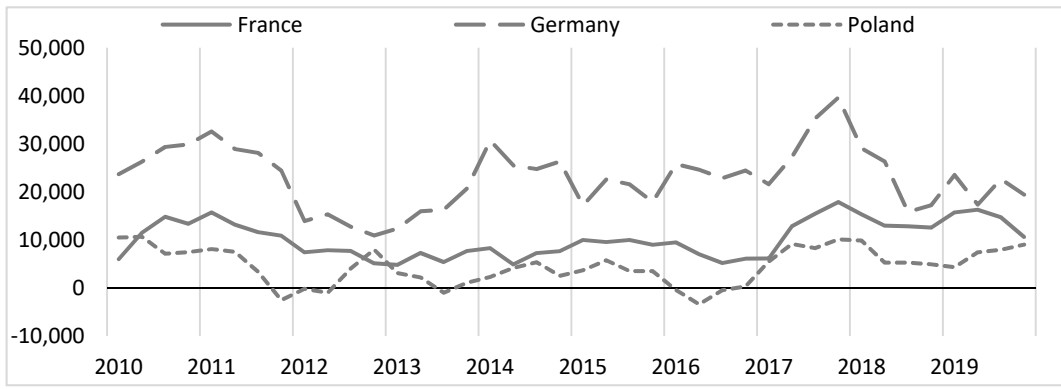

**Figure 8.** Gross value added change (Q/Q of the previous year), own work, source: Eurostat.

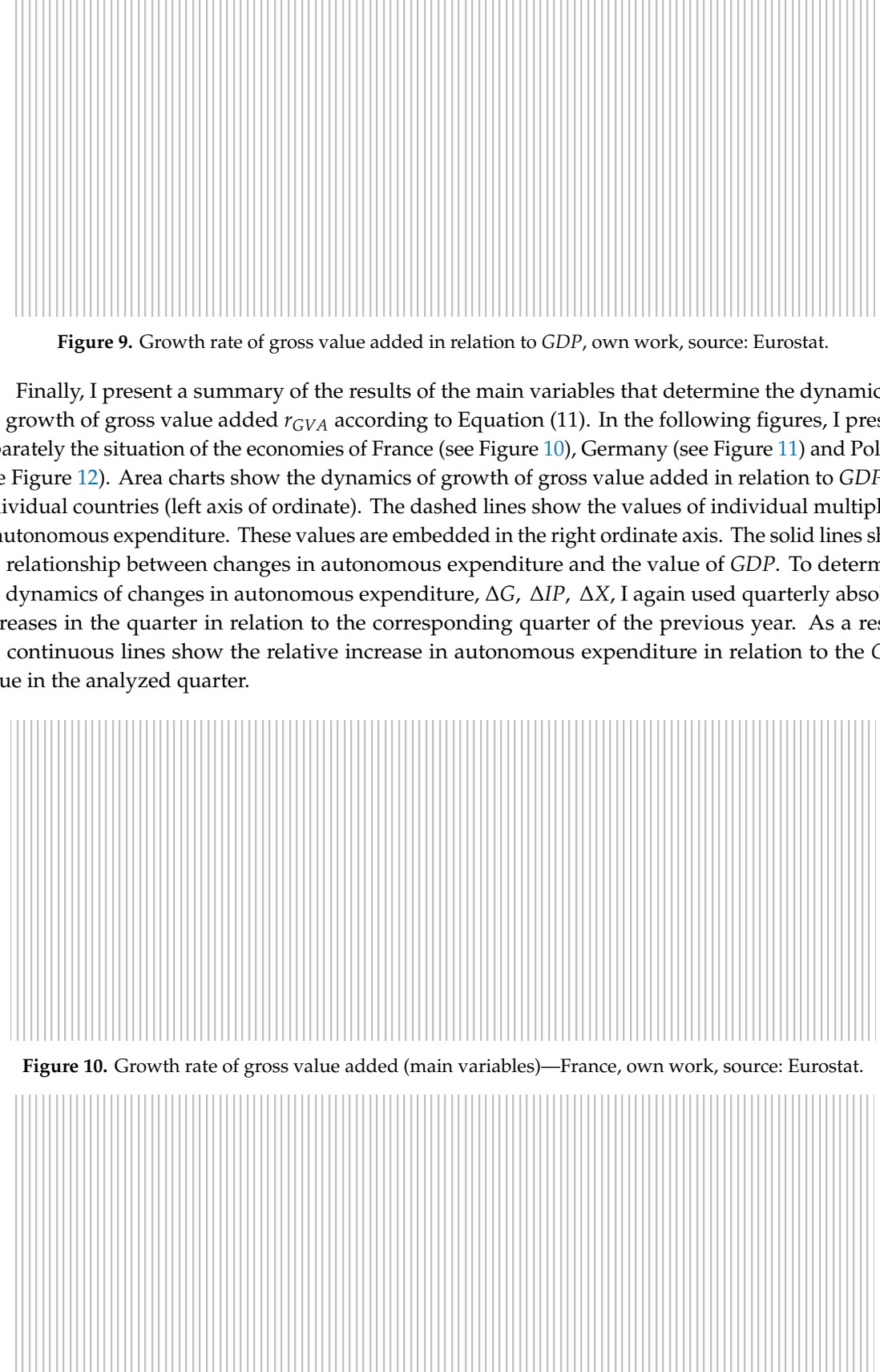

**Figure 9.** Growth rate of gross value added in relation to *GDP*, own work, source: Eurostat.

Finally, I present a summary of the results of the main variables that determine the dynamics of the growth of gross value added $r_{GVA}$ according to Equation (11). In the following figures, I present separately the situation of the economies of France (see Figure 10), Germany (see Figure 11) and Poland (see Figure 12). Area charts show the dynamics of growth of gross value added in relation to *GDP* for individual countries (left axis of ordinate). The dashed lines show the values of individual multipliers of autonomous expenditure. These values are embedded in the right ordinate axis. The solid lines show the relationship between changes in autonomous expenditure and the value of *GDP*. To determine the dynamics of changes in autonomous expenditure, $\Delta G$, $\Delta IP$, $\Delta X$, I again used quarterly absolute increases in the quarter in relation to the corresponding quarter of the previous year. As a result, the continuous lines show the relative increase in autonomous expenditure in relation to the *GDP* value in the analyzed quarter.

**Figure 10.** Growth rate of gross value added (main variables)—France, own work, source: Eurostat.

**Figure 11.** Growth rate of gross value added (main variables)—Germany, own work, source: Eurostat.

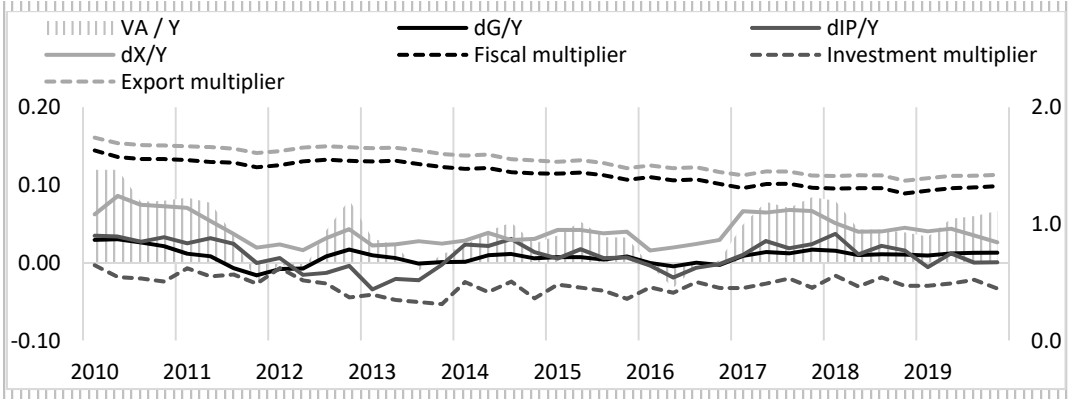

**Figure 12.** Growth rate of gross value added (main variables)—Poland, own work, source: Eurostat.

After empirically verifying the proposed method of calculating the fiscal, investment and export multipliers and relating them to gross value added, some conclusions arose. The change in gross value added in an open economy is determined by two groups of factors. On the one hand, there are injections of government, investment and export expenditure. The increase in these expenditures has a positive effect on the change in the level of gross value added. However, the increase in autonomous expenditure is not equal to the gross value added. This can be seen in Figures 13–15. Here, I present the correlation between the two values. The first is the ratio of the change in autonomous expenditure to *GDP*, i.e., $\frac{\Delta G}{Y}$, $\frac{\Delta IP}{Y}$ and $\frac{\Delta X}{Y}$. The second value is the growth rate of gross value added relative to, $\frac{\Delta GVA}{Y} = r_{GVA}$. It is clearly visible that in each of the analyzed countries, there is a leakage of aggregate demand, which very clearly weakens the importance of autonomous expenditure. These leaks are determined by the value of the fiscal, investment and export multipliers. According to Equation (11), the growth dynamics of *GVA* relative to *Y* depend on changes in individual autonomous expenditures in relation to *Y*. If this were the case, a change in the value of autonomous expenditure would generate exactly the same change in the dynamics of *GVA*. This phenomenon is depicted in Figures 13–15 in the form of an R-squared correlation index. This reveals the impact of changes in autonomous expenditure (ornating axis) on the change in the dynamics of *GVA* (abscissa axis). R-squared correlation rates for France were 0.00 for government expenditure, 0.63 for private investment and 0.56 for exports. In Germany, these figures were 0.11 for government expenditure, 0.50 for private investment and 0.48 for exports. In the Polish economy, the R-squared ratios were 0.82 for government expenditure, 0.47 for private investment and 0.70 for exports. These figures show quite clearly that there are leaks in aggregate demand which 'weaken' the impact of autonomous expenditure on the change in the dynamics of *GVA* compared to *Y*.

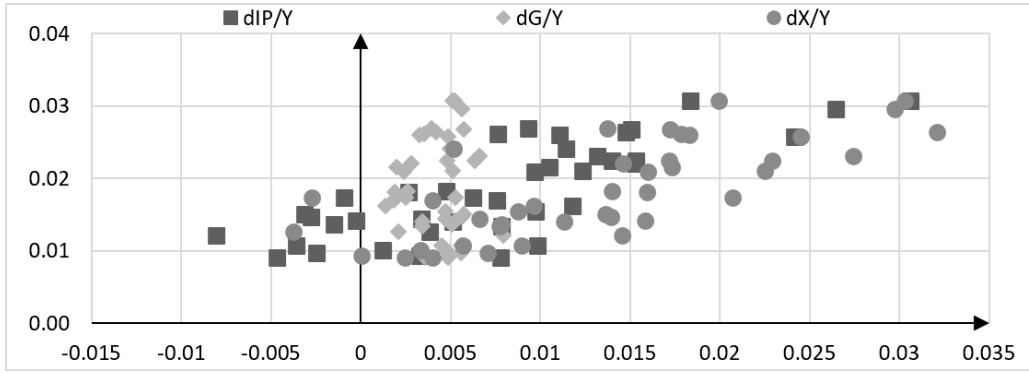

**Figure 13.** Correlation of changes in autonomous expenditure to changes in gross value added—France, own work, source: Eurostat.

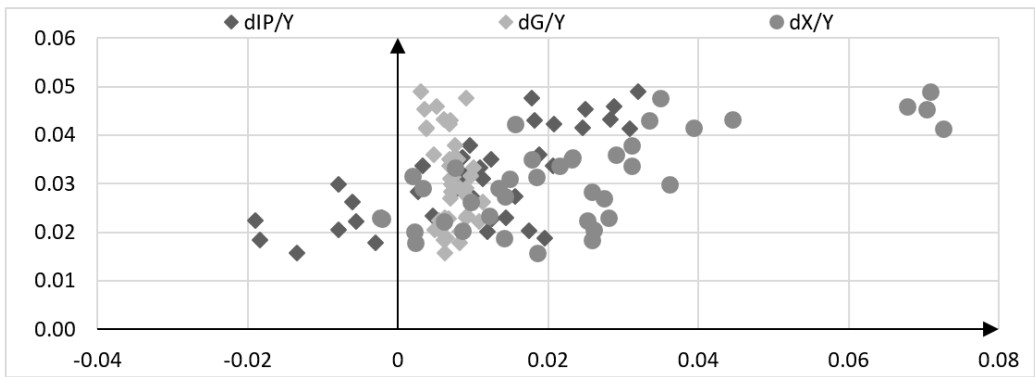

**Figure 14.** Correlation of changes in autonomous expenditure to changes in gross value added—Germany, own work, source: Eurostat.

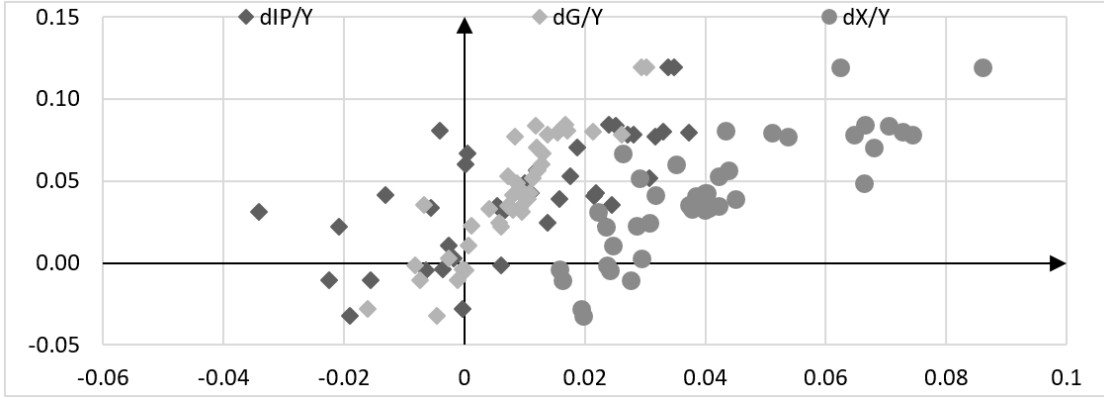

**Figure 15.** Correlation of changes in autonomous expenditure to changes in gross value added—Poland, own work, source: Eurostat.

## 4. Conclusions

In this paper, I set myself two main goals. The first one was an attempt to redefine the mechanism of calculating the fiscal, investment and export multipliers. Using the general economic categories used in the System of National Accounts, I divided the total import into that of intermediate, consumption and capital goods. In addition, I used new coefficients that enabled the division of imported goods into appropriate elements of final production. All this allowed me to estimate the short-term import intensity of all autonomous expenditure. On this basis, it was possible to estimate the fiscal, investment and export multipliers for quarterly periods. The proposed approach makes it possible to analyze aggregate demand leakages in periods shorter than five years. It is in such periods that the most common publication of input–output balance sheets is one of the bases for estimating the autonomous expenditure multipliers in the classical approach.

The second goal was to try to redefine the formula that in the classical approach determines the reaction of *GDP* to the change in autonomous expenditure. I have attempted to change the formula in such a way as to get the answer to the following question: How much does gross value-added change in a capitalist economy as a result of changes in autonomous expenditure? It turned out that after appropriate algebraic transformations, it is possible to obtain an answer to this question. There is still an important mechanism of autonomous expenditure multipliers, which is used in the classical approach. However, there was an additional leakage of aggregate demand in the form of the average tax rate in the economy.

The first conclusion that can be drawn from the above considerations concerns the dynamics of changes in *GVA* value relative to *GDP*. If the multipliers of autonomous expenditure have the form as in the Equation (8), we can interpret their influence as follows: The increase in the value of the

autonomous expenditure multipliers has a positive effect on the change in gross value added. These multipliers, however, are determined by several quantities. First, the level of import intensity should be mentioned. The increase in the import intensity of individual autonomous expenditures reduces their importance. This means that the leakage of aggregate demand in the form of expenditure on the purchase of imported goods intensifies. On the other hand, we are dealing with the import intensity of private consumption. An increase in this value also negatively affects the change in *GVA*. This means that the leakage of aggregate demand, which consists in increasing consumer expenditure allocated to the purchase of imported goods, intensifies. Moreover, the value of the fiscal, investment and export multipliers is determined by the factor *CP*. It determines the propensity for total private consumption. This value, according to Equation (2), depends on the propensity to save and the average taxation of household income. An increase in these two ratios causes a decrease in the value of propensity to private consumption. This, on the other hand, reduces the impact of autonomous expenditure on gross value added. This means that the leakage of aggregate demand increases in the form of a decline in total private consumption in the economy. It may be caused by an increase in savings and/or an increase in the average taxation of household income. Equation (11) reveals one more aggregate demand leak that determines the response of *GVA* to changes in autonomous expenditure. It is the ratio of the average taxation in the economy. Its growth negatively affects this relationship. This means that the leakage of aggregate demand in the form of increased tax burdens in the entire economy intensifies.

Another conclusion concerns the growth dynamics of gross value added in relation to gross domestic production. This indicator shows how this relation changes due to changes in autonomous expenditure. It seems that the ratio of gross value added to *GDP* is very important. Its increase may prove the optimization of the functioning of the domestic economy. The method of calculating the fiscal, investment and export multipliers proposed in this paper, based on the decomposition of imports according to BEC, allows for the analysis of these relationships in short periods. Thus, one can risk a statement that the use of the proposed method may be valuable both for future economic research and for the practice of economic policy.

One more conclusion comes from the considerations presented. It is related to one of the objectives of my research. It turned out that the change in the methodology for calculating import-absorbing indicators makes it possible to calculate fiscal, investment and export multipliers in the short term. The most important value of the proposed method of calculating the multipliers of autonomous expenditure results from short-term observations. This solution enables the analysis and evaluation of economic policy in the short term. Additionally, this method enables the analysis of changes in gross value added in relation to *GDP* in the context of changes in autonomous expenditure and total demand leakages. These issues were the key areas of my research in this manuscript. It can be said that the proposed Equation (11) shows the response of the dynamics of growth of *GVA* to changes in autonomous expenditure in the context of "running" aggregate leakages. It should be emphasized, however, that this reaction concerns a short period of time. On the one hand, it is the source of the value of the proposed method. First of all, in the field of calculating import-intensity rates. On the other hand, however, one cannot forget, for example, about seasonal fluctuations. It could be considered a disadvantage of the proposed solution. However, I believe that this is only a methodological limitation. However, this method is dedicated to short-term analysis. Thus, the long term should also be taken into account for the full picture of the economy. The proposed method also does not answer another question: What happens to leaks in aggregate demand? However, the main goal of this solution is to show the *GVA* dynamics response in the face of these leaks. In this context, there are directions for further research that will complement the proposed method. It seems that studies comparing the calculations of fiscal, investment and export multipliers in short and long periods can be valuable. Comparative studies of the dynamics of *GVA* changes in the face of the situation in short periods and in long periods with seasonal adjustment may also be interesting.

**Funding:** This research was funded by University of Social Sciences.

**Conflicts of Interest:** The author declares no conflict of interest.

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
