# Peer review of "Autonomous Expenditure Multipliers and Gross Value Added"

_jrfm, doi:10.3390/jrfm13090213_

Round 1
Reviewer 1 Report
no comment
Author Response
Thank you kindly for a positive review of the manuscript. I am pleased that he has found recognition from the Reviewer. I made the required adjustments as suggested by the other Reviewers. I marked them by underlining the text.

Reviewer 2 Report
- For a better visibility in databases it is better not to repeat among keywords the words included in the title of the paper.
- Structure of the paper does not meet the editorial requirements, see: Instructions for Authors.
- The research is based on the example of 3 countries. Why were they selected for the analysis? It should be clearly justified. The time range of the research covers the years 2015-2019, i.e. the growth phase of the business cycle. Is it sufficient period to show the usefulness of the proposed method? This should be explained and justified as well. I also suggest to consider extending the time frame of the study in such a way as to show the usefulness of the proposed method at different stages of the business cycle.
- The literature review is very poor, as it contains only 19 items. At the very beginning of the Introduction the author referred to the "analysis of economic literature" and noted that it is one of the most important issues in the development of modern economy. However, any relevant literature was not quoted to prove this fact. It is necessary, especially since in line 68-69 the author states: "I completely reject the assumptions of the mainstream economy, which speaks of the automatic pursuit of a free market economy to a state of equilibrium in full use of production capacity". Obviously, the author can make such an assumption for further analyses, however, it should be clearly justified why he did so. This may be useful for methodological reasons, but the possible implications of this approach should be indicated. Will it be valid as much in a neoliberal economy as in a largely regulated economy? It is important from the point of view of the usefulness of the proposed research method. It would be valuable to open a discussion in this field on the basis of a literature review.
- The value of the proposed method results from the short time intervals of observations, which may be of practical importance in the evaluation of macroeconomic policy in the short run. However, this is also the greatest weakness of the suggested approach. The limitations of this approach should be clearly indicated. First of all, it is necessary to consider how the possible outcomes are affected by both seasonal fluctuations specific to some branches of the economy and by external shocks. It should be also considered what happens to the described "leakages" in the form of taxes, imports and private savings. There are many economic theories explaining their importance in the short and long term. These comments do not concern the formal aspect of the proposed method, but the (equally important) question of its usefulness.
- Figures 13 and 14 showing the correlation coefficients should be presented in the analytical part of the paper, not in the "Conclusion" section (between lines 348 and 349). The "Conclusion" section must be supplemented with conclusions about the usefulness of the proposed method and its limitations (based on a previously conducted discussion). It is also a good practice to indicate some directions for further research.
Author Response
Thank you kindly for all your comments on my manuscript. I made adjustments as suggested. I marked them with an underline of the text.
1. I changed the gross value added to GVA. I find no alternative to the words fiscal multiplier, investment multiplier and export multiplier.
2. I have improved the structure of the article.
3. I chose the Weimar Triangle state. The choice is dictated by the fact that the economies of these three countries are quite different. The aim of this choice of economies is to try to verify the method of calculating the multipliers of autonomous expenditure in different economic circumstances. I extended the timeframe for the study for the period 2010-2019.
4. The rejection of the assumptions of mainstream economics is due to the fact that I fully accept the economics of M. Kalecki and K. Łaski – in this context, it is a theory of total demand. I'm not going to discuss that. The article has a different purpose. I added an explanation in the text. I did not include in the article a review of literature for one reason. I wanted to base my considerations only on the original literature of K. Łaski.
5. I explained this issue in the end of the manuscript.
6. I moved figures 13 and 14 to the analytical part. I have also improved the conclusions.
I hope that these adjustments are satisfactory. Thank you in advance for your help and all your comments. They are very valuable to me.

Reviewer 3 Report
Summary and overall opinion
The manuscript deals with a new approach to computing fiscal, investment, and export multipliers for the short run.
The theoretical construction is a bit thin, but the research question theoretically has value. The discussion is generally coherent but it also tends to be vague or diluted at times, while the manuscript seems updated in terms of references. The manuscript has a certain dose of intrinsic merit but is not publishable as it stands.
Comments and Suggestions
2.1. Main concerns
- The manuscript touches the focus of the journal only tangentially and I think that this contribution would be more suitable in a journal focusing on general economics.
- The research question is in my opinion a little thin and should be heavily discussed in the introduction which is one of the weakest sections of the manuscript. The author fails to identify a clear gap in the literature that the manuscript aims to fill despite discussing the objectives of the manuscript. Moreover, the authors are unsuccessful in detailing the original contributions of the manuscript. In addition to this, the composition technique makes the section rather poor and I strongly advise the authors to rewrite it.
- Despite a fair theoretical construction in section 2, I have several concerns about the relevance of the approach. This should be again heavily discussed in terms of relevance.
- Line 207 introduces a subsection entitled Estimation of autonomous expenditure multipliers. Despite this fact, I could not detect the estimation component as all the ratios reported seem to be the result of simple calculus.
- The Conclusions section could benefit from some upgrades. I would like to see several clear take-aways that derive from the study. Again, upgrades in writing and removal of repetitive sections would be a bonus.
2.2. Minor comments:
- There are many instances of text overlap in several sections.
- Upgrades in the abstract would be a plus
- The manuscript needs consistent upgrades in terms of language and style.
- I do not truly grasp the relevance of the figures included in the conclusions section, nor the logic for that correlation.
Author Response
Thank you kindly for all your comments on my manuscript. I made adjustments as suggested. I marked them with an underline of the text.
1. I justified the research questions in the introduction.
2. I described in the manuscript the most important value of the proposed method for calculating the multipliers of autonomous expenditure. The proposed method allows economic policy to be analysed and evaluated in the short term. In addition, the proposed method of calculating multipliers gives the possibility to calculate them without the balance of inter-branch flows. This applies to the calculation of import-absorbence rates. To date, this is possible once every five years (frequency of publication of this data). Of course, this method will not replace the classic approach according to the theory of total demand. It should be taken as a complement to it.
3. Full agreement on the word 'estimation'. This is my oversight in translating the text. I meant 'calculating'. In the article I accepted the logic that is described by K. Łaski in "Lectures (...)".
4. I have improved the 'Conclusions' sections.
5. I have also clarified the description of the correlation.
I hope that these adjustments are satisfactory. Thank you in advance for your help and all your comments. They are very valuable to me.

Round 2
Reviewer 2 Report
The author took into account the reviewer's comments.
Reviewer 3 Report
Summary and overall opinion
First of all, I would like to congratulate the author for the efforts put into providing a more refined version of the manuscript. The author attempted to tackle several of the concerns I expressed regarding the initial version of the paper, but I consider this revision as unsuccessful. My present review will follow in the line of the comments put forward in my original review, trying to treat them in the same order.
Comments and Suggestions
2.1. Main concerns
Comment 1: I still feel that the manuscript in the current form is out of scope.
Comment 2: Comment 2 is partially tackled in the revision. Even though I acknowledge the effort of the author, the original comment held more substance than what was conducted in the revision.
Comment 3: This is answered to a certain extent in the author's reply and to a lower extent in the manuscript.
Comment 4: I agree with the author that section 3 relies on a calculation procedure and not on an estimation/inference procedure. This was corrected since the first review, however, I am not truly convinced by the relevance of the contribution brought in this approach.
Comment 5: Several useful upgrades have been brought to the conclusions sections.
2.2. Minor comments:
Comment 1 remains unanswered.
Comment 2 remains unanswered.
Comment 3 remains unanswered.